# Preparing Biomass Carbon Fiber Derived from Waste Rabbit Hair as a Carrier of TiO_2_ for Photocatalytic Degradation of Methylene Blue

**DOI:** 10.3390/polym14081593

**Published:** 2022-04-14

**Authors:** Yanfei Chen, Chunyan Wang, Junyan Chen, Shuaishuai Wang, Jingge Ju, Weimin Kang

**Affiliations:** 1State Key Laboratory of Separation Membranes and Membrane Processes, National Center for International Joint Research on Separation Membranes, School of Textile Science and Engineering, Tiangong University, No. 399 BinShuiXi Road, XiQing District, Tianjin 300387, China; isyfchen@163.com (Y.C.); wangchunyan0603@163.com (C.W.); cjy1830011056@163.com (J.C.); 2Shandong Provincial Key Laboratory of Olefin Catalysis and Polymerization, Shandong Chambroad Holding Group Co., Ltd., Economic Development Zone of Boxing County, Binzhou 256500, China; 13608923850@163.com

**Keywords:** biomass carbon, waste rabbit hair, TiO_2_, photocatalytic degradation

## Abstract

In the past few years, biomass carbon materials have gained wide attention from many scholars as TiO_2_ carrier materials to improve photocatalytic activity due to their renewable, green, low-cost, and high-efficiency advantages. In this study, TiO_2_/carbonized waste rabbit fibers (TiO_2_/CRFs) nanocomposites with the hierarchical microporous/mesoporous structure were fabricated by a combination of carbonization, immersion, and calcination methods using tetrabutyl titanate as the titanium source and waste rabbit hair as the carbon source. The properties and catalytic activity of TiO_2_/CRFs composite were evaluated based on several characterization techniques and methylene blue (MB) photodegradation studies. The results showed that the degradation of MB by TiO_2_/CRFs could reach 98.1% after 80 min of solar irradiation. Moreover, TiO_2_/CRFs still maintained high photocatalytic activity after five cycles of degradation tests, exhibiting good stability and reusability. The improved photocatalytic performance of TiO_2_/CRFs materials is attributed to the natural carbon and nitrogen element doping of TiO_2_/CRFs and its morphology, which reduces the compounding of photogenerated electron-hole pairs and narrows the TiO_2_ band gap, while the multiple reflections of visible light in the pore channels enhance the visible light absorption of the materials. Furthermore, the large specific surface area provides abundant reaction sites for adsorbed reactants. This paper provides the experimental basis for the application of waste rabbit biomass carbon composites in photocatalytic degradation field.

## 1. Introduction

Textile wastewaters contain a large variety of dyes and chemical substances which cause serious environmental problems that affect human health and aquatic life. Therefore, effective treatment is necessary before emissions. Photocatalysis is considered to be one of the most effective techniques for treating wastewater laden with organic contaminants [1]. The common photocatalysts are mostly metal oxides, such as TiO_2_ [2,3], WO_3_ [4], ZnO [5], SnO_2_ [6,7], BiVO_4_ [8,9], Bi_2_O_3_ [10,11], and ZrO_2_ [12]. Among them, TiO_2_ has attracted wide attention because of its strong oxidation ability, high activity, long-term stability, strong ability to absorb ultraviolet rays, no irritation, low cost, and relatively non-toxicity [13,14]. Especially, anatase phase TiO_2_ has the advantage of high adsorption performance, strong affinity with organic molecules, and low electron-hole recombination rate, which present excellent application performance in the field of photocatalysis [15,16,17]. Nevertheless, TiO_2_ as a highly efficient photocatalyst still has two major defects limiting its large-scale application so far. First, TiO_2_ particles are too small to be separated from the water treatment system. Second, TiO_2_ has a wide bandgap, so ultraviolet light is needed in the photocatalytic reaction, which limits the possibility of using solar energy. To overcome these shortcomings, some scholars started attaching TiO_2_ to the body of carbon materials to prepare photocatalytic materials. On the one hand, carbon materials as carriers to TiO_2_ can recycle in the photocatalytic reaction. On the other hand, carbon materials can also effectively improve the performance of TiO_2_ by narrowing the band gap [18,19,20]. For example, Bin Wang et al. [21] loaded titanium dioxide on carbonized cotton fibers. The results implied that the TiO_2_ nanorods were successfully attached to carbonized cotton fibers. At the same time, the combination of titanium dioxide and carbonized cotton fibers can narrow the bandgap of titanium dioxide, and provide a sustainable and low-cost method to load nanostructured TiO_2_ on carbonized cotton fibers for environmental protection. Cheng Gang et al. [22] researched TiO_2_@RGO, TiO_2_@CNTs, and TiO_2_@C (glucose carbon) hybrid composites. These hybrid materials show enhanced visible light absorption and RhB-dye removal capability via adsorption and photocatalysis with their efficiency generally increasing with carbon content. Nguyen et al. [23] studied the photodegradation of methylene orange and methylene blue dyes catalyzed by palladium doped titanium dioxide (Pd-TiO_2_). The highest decolorization and mineralization were obtained with 0.5 wt.% Pd-TiO_2_ for single dyes and 0.75 wt.% Pd-TiO_2_ in the case of dye mixture. Orooji et al. [24] used tungstophosphoric acid (HPW) to locally modify gold nanoparticles on the surface of TiO_2_. The prepared nanocomposites have good multifunctional photocatalysis for the degradation of nitrobenzene under visible light irradiation and can be reused.

In the application of carbon materials, biomass-derived carbon materials have attracted widespread attention due to their advantages of renewability, ease of processing, controllable surface properties, and relatively low cost. Especially, Rabbit hair fiber with hollow structure is a sustainable protein biomass material, which could be an excellent carrier after carbonization. Large amounts of trimmings are produced in the textile processing and products of rabbit hair. According to statistics, dozens of tons of waste rabbit hair will be directly discarded in landfills every year. These rabbit hair wastes are animal protein fibers, which are quite rich in carbon and nitrogen elements; it is necessary to make full use of these. In this study, waste rabbit hair was used as a bio-carbon material. On one hand, it provides carbon and nitrogen sources for the catalysts, which means nitrogen can be introduced into the composites based on the introduction of carbon doping. It can further reduce the bandgap energy while broadening the light absorption range, and eventually exhibit stronger photocatalytic performance than pure TiO_2_ [25,26,27]. For example, Guidong Yang prepared nitrogen-doped TiO_2_ nitrogen materials. It was shown that the nitrogen-doped TiO_2_ catalysts enhanced the absorption in the visible region and showed higher activity for the photocatalytic degradation of model dyes (MB) [28]. On the other hand, rabbit hair fiber has a natural hollow structure, and the surface of the fiber has a scale layer so it is more conducive to the loading of other materials, has a larger specific surface area, more thin layers, and a stronger visible light absorption capacity [29]. The above characteristics can provide better adsorption, more convenient charge transfer channels, and catalytic activity of the material. In addition, the biocarbon fiber facilitates the migration of photogenerated charges, which greatly improves the separation efficiency of photogenerated electron-hole pairs and the performance of photocatalytic degradation of organic pollutants by the catalyst. Recycling these waste rabbit hairs has important significance in the field of photocatalysis, and making effective use of this will give play to potential value.

In this work, biomass protein carbon fibers (carbonized waste rabbit hair fibers) worked as a carrier for TiO_2_ and carbon and nitrogen doped TiO_2_/CRFs composites with hierarchical structure by a simple impregnation and calcination method, which were for the photocatalytic degradation of methylene blue. The photocatalytic performance for methylene blue degradation of the TiO_2_/CRFs composites was discussed under the conditions of different starting TiO_2_: CRFs mass ratios, photocatalyst dosage, reaction time, and initial concentration of methylene blue. The possibility of material recovery and reuse was tested through cyclic experiments. The prepared material has a large specific surface area, so that the anatase titanium dioxide nanoparticles are uniformly filled on the scale structure of CRFs, which improves the adsorption capacity of the dye, provides more reaction sites, and solves the problem of difficult TiO_2_ recycling. In addition, the natural carbon and nitrogen doping makes the TiO_2_/CRFs composites play an important role in reducing the band gap width. More importantly, it also provides a new material with excellent performance for TiO_2_/C family, and also has wide application prospects in energy storage and adsorption and other aspects.

## 2. Materials and Methods

### 2.1. Chemicals and Materials

The glacial acetic acid was bought from Tianjin Yongsheng Fine Chemical Co., Ltd. (Tianjin, China). The tetrabutyl titanate was purchased from Tianjin Kermel Chemical Reagent Co., Ltd. (Tianjin, China). The anhydrous ethanol was purchased from Tianjin Fengchuan Chemical Reagent Technology Co., Ltd. (Tianjin, China). The methylene blue (MB) was bought from Tianjin Hengxing Chemical Reagent Manufacturing Co., Ltd. (Tianjin, China). In addition, the rabbit hair was bought from a local farm in the Shandong province of China. All of the chemical reagents used have not been further purified.

### 2.2. Preparation of CRFs Carbon Carrier

In order to remove impurities, the rabbit hair was washed with distilled water three times and then dried for 12 h at 80 °C. Subsequently, the rabbit fiber was pretreated in the muffle furnace at 300 °C with a rate of 2 °C min^−1^ in an open-air atmosphere for 90 min to maintain its hollow shape during the carbonization process. Finally, it was carbonized for 1 h at 600 °C with a rate of 2 °C min^−1^ under a nitrogen atmosphere in the tube furnace to obtain carbonized rabbit hair fiber (CRF).

### 2.3. Fabrication of TiO_2_/Carbonized Rabbit Hair Fibers Composites (TiO_2_/CRFs)

The impregnated solution was prepared by blending tetrabutyl titanate, glacial acetic acid, and anhydrous ethanol with a mass ratio of 1:2:4. Subsequently, the CRFs were added into the above-impregnated solution with different mass ratios of tetrabutyl titanate (in impregnated solution) and CRF (3:1, 5:1, and 7:1). After immersion for 4 h, the solution with CRFs were transferred to a vacuum oven (70 °C) for drying. Finally, the dried samples were calcinated to 450 °C for 1 h in a muffle furnace with a heating rate of 5 °C min^−1^. The obtained TiO_2_/CRFs composites were labeled as TiO_2_/CRFs = 3:1, TiO_2_/CRFs = 5:1, and TiO_2_/CRFs = 7:1 according to different mass ratios during the immersion process, respectively. The schematic of the formation of TiO_2_/CRFs composite materials was distinctly presented in Figure 1.

### 2.4. Characterization

The surface morphologies of CRFs and TiO_2_/CRFs were examined with a field emission scanning electron microscope (FE-SEM, Gemini SEM500, Oberkochen, Germany) after sputtering a thin gold layer. Meanwhile, the elemental mapping was obtained by energy disperse spectroscopy (EDS) connected to the FE-SEM. The crystalline phases of the samples were confirmed with X-ray diffraction (XRD) method using a Bruker AXS D8 Discover machine, and the diffraction angle 2θ was recorded from 10° to 80° with Ni-filtered CuKα radiation (λ = 0.154 nm). Moreover, the thermal properties were investigated by thermogravimetric analysis (TGA, NSK, TG/DTG 6300) under the air atmosphere at a heating rate of 10 °C min^−1^. The specific surface areas and pore size distribution of the synthesized samples were measured via physical adsorption at 77 K (BET, Quantachrome Instruments Autosorb-iQ, Boynton Beach, Florida, USA) following the nitrogen adsorption-desorption isotherms. In addition, the surface elemental composition of TiO_2_/CRFs was analyzed using an X-ray photoelectron spectroscopy (XPS) (K-al phaX, Thermo Fisher Co., Waltham, MA, USA). Fourier transform infrared spectra were carried out by diffused reflectance using a Fourier transform infrared spectrometer (FTIR, Nicolet iS50, Thermo Fisher Co., Ltd., Waltham, MA, USA). The UV-visible diffuse reflectance spectra were measured by a UV-visible diffuse reflectance spectrophotometer (Mapada UV-1800, Shanghai, China), and the spectra were recorded at room temperature in the air from 200 to 800 nm.

### 2.5. Evaluation of Photocatalytic Activity

Methylene blue (MB) is a common dye, which is extensively used in various industrial applications. In this study, the performance of the catalyst was tested by photocatalytic degradation of MB dyes. An appropriate amount (10 mg L^−1^) of MB was taken to simulate the wastewater and diluted with different multiples to obtain the MB solution with concentrations of 1, 2, 4, 6, 8, and 10 mg L^−1^, respectively. The standard working curve of MB solution was obtained by measuring the absorbance of MB solution at different concentrations, which was shown in Figure 2. After numerical fitting, the concentration of MB solution C (mg L^−1^) is linearly related to its absorbance A within a certain range, which can be approximately expressed as: A = 0.15836 C (R^2^ = 0.99935). Subsequently, an appropriate amount of photocatalyst was placed in 60 mL MB solution. First, they were stirred in a dark environment (300 r min^−1^) for 30 min to achieve adsorption equilibrium. Then, the beaker is exposed to ultraviolet light generated by an 18w UV lamp (the main wave peak is at 365 nm and the UV lamp is at 30 cm of the beaker) and solar light. Samples are taken at regular intervals to measure the absorbance at different times.

## 3. Results

### 3.1. Morphologies and Chemical Properties

The morphologies for raw rabbit hair and carbonized rabbit hair fibers were presented in Figure 3a,b, respectively. It can be observed from Figure 3b that the surface of carbonized rabbit hair fiber has a rough cuticle layer, which is attribute to the special scale structure of the rabbit hair. In addition, the rough scale layer exposes more binding sites, which is more conducive to the loading of TiO_2_ particles. The SEM images of TiO_2_/CRFs with different doping contents of TiO_2_ nanoparticles are shown in Figure 3c–f. It can be seen that the number of TiO_2_ nanoparticles increases as the amount of immersion solution increases. When the mass ratio of TiO_2_ and CRFs is 3:1 (Figure 3c), there is only a small amount of TiO_2_ nanoparticles attached to the CRFs. When the mass ratio of TiO_2_/CRFs is 5:1 (Figure 3d,e), more TiO_2_ nanoparticles are distributed in the scale layer of CRF, forming a uniform layered structure [30]. However, as the ratio continues to grow, when the mass ratio of TiO_2_/CRFs reaches 7:1, TiO_2_ nanoparticles seem to be agglomerated (Figure 3f). By contrast, when the mass ratio of TiO_2_/CRFs is 5:1, the distribution of TiO_2_ nanoparticles is more uniform, which would provide more active sites in the catalysis process.

Figure 4 presents the X-ray energy dispersive spectrum of CRFs and TiO_2_/CRFs (TiO_2_/CRFs = 5:1). It indicates that there are five elements, including C, N, O, S and Ti, that exist in the composite material. Among them, C, N and S element derives from carbonized rabbit hair fiber as well as Ti and O coming from loaded nanoparticles.

Figure 5a presents XRD patterns of CRFs, TiO_2_, and TiO_2_/CRFs. The XRD pattern corresponding to CRFs shows a broad peak (2θ = 26°). This angle corresponds to the graphite phase of carbon, which proves the formation of the graphite phase during carbonization [31]. In addition, it can be observed that the prepared TiO_2_/CRFs = 5:1 has the same XRD pattern peaks with pure anatase TiO_2_ (JCPDS No.21-1272) [32]. It is known that TiO_2_ has three crystal forms: anatase, brookite, and rutile. Among them, anatase TiO_2_ has a low dielectric constant, low mass density, and high electron mobility, and the oxygen vacancies are larger than brookite crystals and rutile crystals, so anatase crystals have the highest catalytic activity [33]. In short, this article successfully synthesized a TiO_2_/CRFs composite catalyst with anatase crystal form, indicating its higher photocatalytic activity.

To confirm the contents of TiO_2_ in TiO_2_/CRFs = 5:1 composite material, the TG/DTG curves of TiO_2_/CRFs were presented in Figure 5b. The degradation occurred within a wider temperature range and showed two well separated processes in the DTG curves. The first thermal mass loss in the heating process occurred between 20 and 100 °C, corresponding to the evaporation of water molecules. The second mass loss occurred when the temperature was between 500 and 600 °C, which corresponds to the thermal decomposition of CRFs. At the same time, due to the high melting point, the remaining material is TiO_2_. The results showed that the high loading mass ratio of TiO_2_ into the TiO_2_/CRFs was up to 57.31%, which was benefited from the cuticle layer and pore structure of CRFs (as shown in Figure 3d).

The low-temperature nitrogen adsorption-desorption isotherm curves of TiO_2_/CRFs samples with different mass ratios were illustrated in Figure 5c. According to the standard IUPAC classification, the low-temperature nitrogen adsorption isotherm patterns of all samples belong to a combination of type I and IV isotherms, indicating that the pore structure of TiO_2_/CRFs materials has a variable distribution and a multi-scale pore structure [34]. The adsorption and desorption curves in Figure 5c are inconsistent, and hysteresis loops can be observed. According to the standard IUPAC, TiO_2_/CRF presents H4-type hysteresis loops, which are mostly found in solids with narrow fissure pores, distinguished from particle stacking. It is a kind of hole similar to that produced by the layered structure, in accordance with the previous SEM image showing that TiO_2_ particles are mostly stacked at the scale layer of CRFs, showing a layered distribution. Combined with the pore size distribution plots of the TiO_2_/CRFs samples in Figure 5d, TiO_2_/CRFs have more micropores (<2 nm) compared with CRFs, mainly because a small amount of TiO_2_ nanoparticle loading fills and plugs part of the mesopores (2–50 nm) in CRFs, forming a hierarchical micropore/mesopore structure. With the increase of TiO_2_ nanoparticles up to the amount of CRFs/TiO_2_ = 1:7, the number of mesopores increases, which may be due to the excessive accumulation of TiO_2_ nanoparticles in the fiber scale layer and the formation of larger mesopores between the particles. The hierarchical microporous/mesoporous structure endows TiO_2_/CRFs materials with higher light-trapping ability, shorter transport distance of photo-excited electron/hole, and higher specific surface area. Abundant pore channels afford a more effective transport path and facilitate the spread of molecules in the reaction process. Additionally, a porous structure can also promote the migration of electrons, which has been shown to suppress the carrier recombination [35].

The XPS spectra of TiO_2_/CRFs were presented in Figure 6, which presented that it predominantly contained N, Ti, C, and O elements [28]. The Ti 2p characteristic peaks observed at 459.0 and 464.8 eV were attributed to the existence of Ti^4+^ in TiO_2_/CRFs [32]. The high-resolution XPS spectrum of C 1s was shown in Figure 7a. Many peaks within the range of 282 eV to around 292 eV may be ascribed to the C–C (284.5 eV), C=C (285.2 eV), C–O (286 eV), and C=O (289.1 eV) bonds. Compared with CRFs, the peak area of C=C decreases, and the peak area of C-C increases in TiO_2_/CRFs, which indicates that C=C sp2 hybridization (graphite state) is converted to C-C sp3 hybridization. It is shown that the heteroatoms (including O, Ti, etc.) are bonded in the form of chemical bonds on carbon fibers instead of simple intermolecular binding [36]. The spectrum of O 1s is given in Figure 7b. The binding energy (BE) values of the individual components are 530.5 (Ti^4+^-O), 531.6 (Ti^3+^-O), and 532.76 eV (OH^−^). Compared with pure TiO_2_, the appearance of O-Ti^3+^ in TiO_2_/CRFs proves that the defect concentration increased and the TiO_2_/CRFs display negative O-Ti^4+^ BE shift in the O 1s level, which further confirms the formation of O–Ti–N bonds [25]. The N1s XPS spectra for TiO_2_/CRFs are shown in Figure 7c. The N element comes from the natural rabbit hair, and it is successfully doped into TiO_2_ during prepared processes. The broad peak can be fitted by four peaks at 398.2, 399.5, 400.7, and 397.0 eV, suggesting four independent environments for N within TiO_2_/CRFs. Regarding CRFs materials, three types of N doping in carbon materials can be determined, namely Pyridinic N, Pyrrolic N, and graphitic N [37]. On the one hand, TiO_2_/CRFs have an N-Ti peak than CRFs, which proves the doping of N in TiO_2_ crystals and the chemical bonding of TiO_2_ crystal with N element in CRFs. On the other hand, the content of pyrrole N in TiO_2_/CRFs increased significantly, which is presumed to be caused by the etching effect of glacial acetic acid on CRFs during the co-heating process. This also means that the N element changes from the state of connecting carbon atoms (Pyridinic and graphitic Nitrogen) to the transition trend of connected heteroatom states (Pyrrolic Nitrogen). This transformation may be beneficial to generate more active sites that can be combined with Ti.

Figure 7d shows the FTIR spectra of CRFs and TiO_2_/CRFs. The two samples have similar spectra with the strong and broad absorption bands around 3400 cm^−1^ attributed to surface hydroxyl groups and absorbed water molecules [38]. The positions of 2965 cm^−1^ and 2863 cm^−1^ are -CH_2_- stretching vibration bands [22]. The strong band located at 500~700 cm^−1^ in TiO_2_/CRFs attribute to Ti–O stretching and Ti–O–Ti bridging stretching modes [28]. The peak at around ~1630 cm^−1^ corresponds to bending vibrations of O–H and N–H [28], and the band at around 1474 cm^−1^ is attributed to the vibrations of the Ti–N bond [25]. The appearance of the N–Ti bond in the samples suggests that the N species have been incorporated into the TiO_2_ lattice. This finding is in accordance with the XPS result previously discussed.

### 3.2. Photocatalytic Activity Performance

Figure 8a shows the photocatalytic degradation activity of MB with different loads of TiO_2_/CRFs. The as-prepared TiO_2_/CRFs are systematically evaluated under the condition that the mass concentration of MB was 10 mg L^−1^ (60 mL) and the sample amount of TiO_2_/CRFs was 50 mg. It is known from the literature that in the dark adsorption process, bare TiO_2_ basically has no adsorption effect on MB [39,40]. In the case of dark reaction for the first 30 min, TiO_2_/CRFs = 3:1 has the best adsorption performance, which is because the specific surface area of TiO_2_/CRFs = 3:1 is much larger than those of TiO_2_/CRFs = 5:1 and TiO_2_/CRFs = 7:1 (as listed in Table 1). In the following reaction under UV light, it is obvious that the photocatalytic degradation efficiency of MB solution first improves as the ratio of TiO_2_ and CRFs increasing. The reason could be explained that the increase of TiO_2_ load brings more active reaction sites. As the ratio of the two further increases, the degradation efficiency of the sample to MB solution decreases. This is because the TiO_2_ load continues to increase but the dispersibility is relatively poor, which reduces the effective adsorption of MB molecules and results in the reduction of the degradation efficiency. When the TiO_2_/CRFs = 5:1, the degradation efficiency of the sample to MB solution reached the maximum.

To determine the photocatalytic reaction rate, an attempt is made to fit the data using some common kinetic equations, which are shown in Figure 8b. The first-order kinetics is then confirmed by making a linear plot $lnC/C_{0}$ against time. The kinetic equation could be expressed as follows [41,42]:(1)$$ln\frac{C}{C_{0}}=kt$$
where $C_{0}$ is the concentration of the reactant before illumination (mg L^−1^); *C* is the concentration of the reactant after a certain illumination period *t* (mg L^−1^); *k* is the first-order rate constant (min^−1^) and *t* is the illumination time (min).

The kinetics of MB photoreduction with different samples are illustrated in Figure 8b, respectively. It is clearly seen that the photooxidation of MB follows a first-order kinetics equation. Figure 8c shows the influence of TiO_2_/CRFs = 5:1 dosage on the degradation performance of MB solution under the conditions of UV-light irradiation for 60 min and MB solution concentration of 10 mg L^−1^. It can be seen that the degradation rate of MB solution increases with the increase of TiO_2_/CRFs dosage. Specifically, when the dosage of TiO_2_/CRFs was 0.41 g L^−1^, 1.25 g L^−1^ and 1.46 g L^−1^, the degradation rates of MB solution were 41.9%, 97.9% and 98.3% respectively. However, when the amount of TiO_2_/CRFs is greater than 1.25 g L^−1^, the degradation rate of MB solution does not increase significantly. This is because increasing the amount of TiO_2_/CRFs at the beginning can increase the number of surface activity sites of TiO_2_ photocatalyst, thus increasing the degradation rate of MB solution [43]. However, due to the limited area of TiO_2_/CRFs photocatalyst exposed to UV-light, and the excessive TiO_2_/CRFs photocatalyst will block UV light, increasing the amount of TiO_2_/CRFs will not significantly improve the degradation rate. Therefore, in the photocatalytic experiment (60 mL 10 mg L^−1^ MB), the optimal amount of TiO_2_/CRFs photocatalyst is 1.25 g L^−1^, and the degradation rate can reach 97.9%. Compared with bare TiO_2_, the degradation rate of MB solution is only 85.35% when the UV lamp is irradiated within 60 min. [40]. Figure 8d shows the photocatalytic degradation activity with different MB concentrations under the condition that the sample of TiO_2_/CRFs = 5:1 was 1.25 g L^−1^ and the volume of MB was 60 mL. The degradation rates of the samples after 60 min of irradiation were 98.0% (5 mg L^−1^ MB), 97.9% (10 mg L^−1^ MB), 88.8% (15 mg L^−1^ MB) and 50.9% (20 mg L^−1^ MB), respectively. By comparison, the photocatalyst efficiency is similar when MB concentrations are 5 mg L^−1^ and 10 mg L^−1^, so the MB solution with a concentration of 10 mg L^−1^ is an appropriate choice.

The above three experiments (Figure 6c,d and Figure 8a) were designed to explore the proportion, dosage, and appropriate MB concentration of TiO_2_/CRFs composite materials under more stable conditions. However, in the face of industrial application, it is very important that dyes can be directly degraded by visible light irradiation. Therefore, the solar photocatalytic degradation activity of TiO_2_/CRFs has been shown in Figure 9a. Moreover, to further compare the advantages of rabbit fur as a carrier, Figure 9a also shows the comparison of photocatalytic degradation activities of TiO_2_/CRFs and TiO_2_/Cotton fibers under the condition of sunlight. The degradation rates of the two samples after 80 min of sunlight irradiation are 98.1% and 43.3%, respectively. According to a large amount of literature that has focused on the photocatalytic activity mechanism of N-doped TiO_2_, it is generally believed that N doping in TiO_2_ lattice changes the electronic band structure of TiO_2_, resulting in the formation of a new substituted N 2p band above the O 2p valence band, which reduces the band gap of TiO_2_ and transfers the optical absorption to the visible region [26,28,44]. In Figure 9a, the photocatalytic degradation activity of TiO_2_/CRFs in sunlight is significantly stronger than that of TiO_2_/Cotton fibers. It can be determined that this phenomenon is due to the N-doping of TiO_2_/CRFs, which decreased band-gap energy of the catalyst and thus enhances the solar energy absorption. Figure 9b shows the sunlight absorption curve of the TiO_2_/CRFs composite. It can be seen that the maximum absorbance value of MB is significantly reduced, and nearly all MB is degraded after 80 min of sunlight irradiation, indicating TiO_2_/CRFs have a significant photocatalytic degradation effect on MB.

Figure 10 shows the possible degradation mechanism of MB by TiO_2_/CRFs photocatalyst. After light irradiation, TiO_2_ nanoparticles generate electron-hole pairs after absorbing light with energy equal to or greater than the bandgap energy of TiO_2_ nanoparticles. These light-generated charge carriers are responsible for the photocatalytic degradation of MB by TiO_2_ nanoparticles. However, the rapid recombination of photogenerated electron-hole pairs leads to the low photocatalytic activity of TiO_2_ nanoparticles. The combination of TiO_2_ nanoparticles and CRFs nanomaterials improves photocatalytic efficiency mainly through two mechanisms, including greater adsorption on MB and feasible charge separation and transport. According to a series of carbonaceous–TiO_2_ composites with unique morphologies shown in Ref. [45], the carbonaceous nanomaterials can significantly increase the dye concentration on the photocatalyst surface with good adsorption properties. The results indicated that the bulk adsorption performance of dye molecules on the photocatalyst surface is a key factor to improve photocatalytic activity. In addition, the carbonaceous nanomaterials can play the central role of electron reservoir to obtain electrons from the electron-hole pairs of TiO_2_, thus improving the charge separation efficiency of TiO_2_. As seen in Figure 10, when TiO_2_ nanoparticles are irradiated by light, they may generate electrons in the conduction band (CB) and holes in the valence band (VB), and may also form reactive substances such as OH and O^2−^ [46,47]. Therefore, the generated cavities and oxidizing species will break down the MB into CO_2_ and water by hydroxylation. Furthermore, it is known that carbonaceous nanomaterials can act as electron traps, allowing the rapid transfer of a fraction of photogenerated electrons from TiO_2_ nanoparticles CB to carbonaceous nanomaterials and the effective separation of photogenerated charge carriers [45]. Thus, electrons are transferred to the surface where they react with adsorbed oxygen (O_2_) to produce highly reactive superoxide radical anions (O^2−^), while cavities can oxidize H_2_O to OH radicals [48,49,50]. The stability of the photocatalyst under solar light irradiation is an essential indicator of the performance of photocatalytic degradation. TiO_2_/CRFs were centrifuged and reclaimed for MB degradation at a similar dye concentration. The regeneration analysis of the degradation of MB dye by TiO_2_/CRFs under natural solar light irradiation is exhibited in Figure 9c. From the results of regeneration cycles, the degradation efficiency of TiO_2_/CRFs of MB was reduced by 3.9% compared with the first cycle, indicating that the catalyst has a good persistent catalytic ability. Figure 11 shows before and after photocatalytic degradation of methylene blue solution. Table 2 shows the comparison of the results of the TiO_2_/CRFs prepared in this study with other titanium dioxide composite photocatalysts in the literature for the degradation of MB, and it was found that the TiO_2_/CRFs prepared in this study had a significant advantage in degrading MB, maintaining a high level of effectiveness.

## 4. Conclusions

In this study, a novel TiO_2_/CRFs nanocomposite photocatalyst was designed by combining carbonization, impregnation, and calcination processes with improved photocatalytic performance and excellent stability. TiO_2_/CRFs are naturally doped with carbon and nitrogen elements from waste rabbit hair, which effectively reduce the compounding of photogenerated electron-hole pairs and the band gap of TiO_2_, and the absorption of visible light is enhanced, thereby greatly improving the photocatalytic activity of TiO_2_. Its large specific surface area and special hierarchical micropore/mesopore structure provide abundant active sites for the adsorption of reactants, and the multiple reflections in the pore structure enhance the light-trapping ability of the material. Thereby, efficient charge separation and strong light-harvesting capability are achieved, realizing the ultimate goal of the removal of methylene blue under visible light. The experiments showed that the photocatalytic degradation of MB solution by TiO_2_/CRFs could reach 98.1% after 80 min solar irradiation at the initial concentration of MB 10 mg/L, the ratio of TiO_2_/CRFs 5:1, and material dosage 1.25 g/L. In addition, the TiO_2_/CRFs photocatalyst can be easily recovered with only 3.9% loss of photocatalytic activity after five cycles. Further systematic studies on TiO_2_ doping with C and N will be conducted in the future. In consideration of the low cost and waste reuse of rabbit hair, as well as the no-toxic and high photocatalytic performance of TiO_2_, this work undoubtedly provides an economic and environmental strategy for the preparation of biochar/TiO_2_ composite photocatalysts. It is competitive among related materials and provides experimental basis for the subsequent photocatalytic research of carbon fiber and TiO_2_ composite nanomaterials.

## Figures and Tables

**Figure 1 polymers-14-01593-f001:**
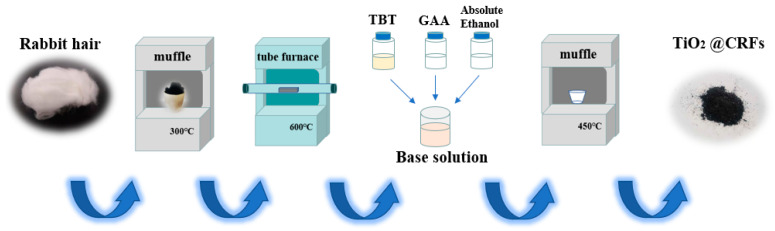
Schematic of the formation of TiO_2_/CRFs composite materials.

**Figure 2 polymers-14-01593-f002:**
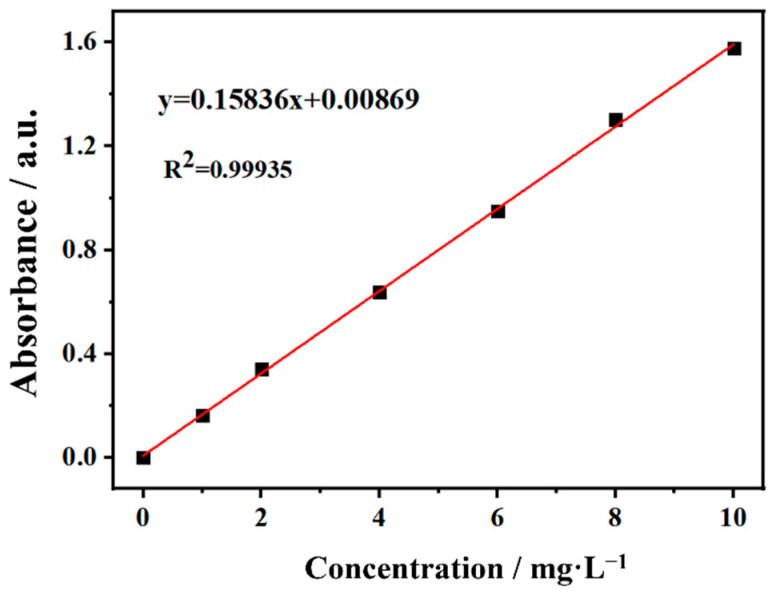
The standard working curve of MB solution.

**Figure 3 polymers-14-01593-f003:**
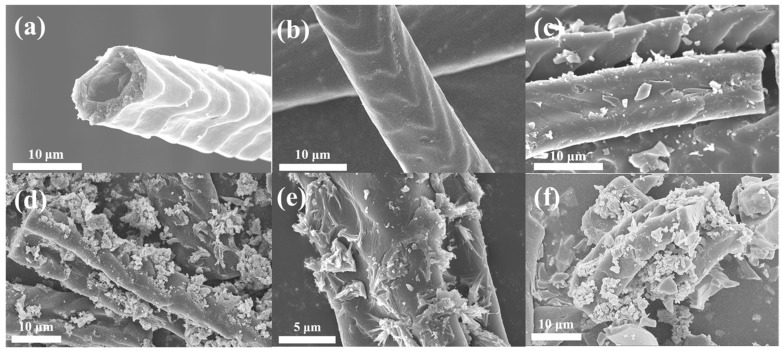
SEM images of (**a**) rabbit hair fiber, (**b**) carbonized rabbit hair fiber, (**c**) TiO_2_/CRFs = 3:1, (**d**) TiO_2_/CRFs = 5:1, (**e**) TiO_2_/CRFs = 5:1 (partial enlarged detail) and (**f**) TiO_2_/CRFs = 7:1.

**Figure 4 polymers-14-01593-f004:**
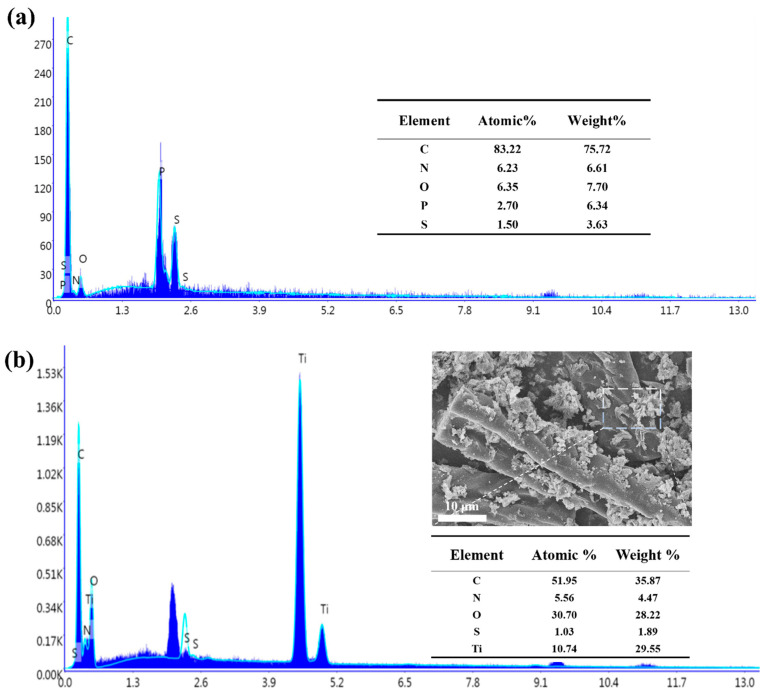
X-ray energy dispersive spectrum of (**a**) CRFs and (**b**) TiO_2_/CRFs (TiO_2_/CRFs = 5:1).

**Figure 5 polymers-14-01593-f005:**
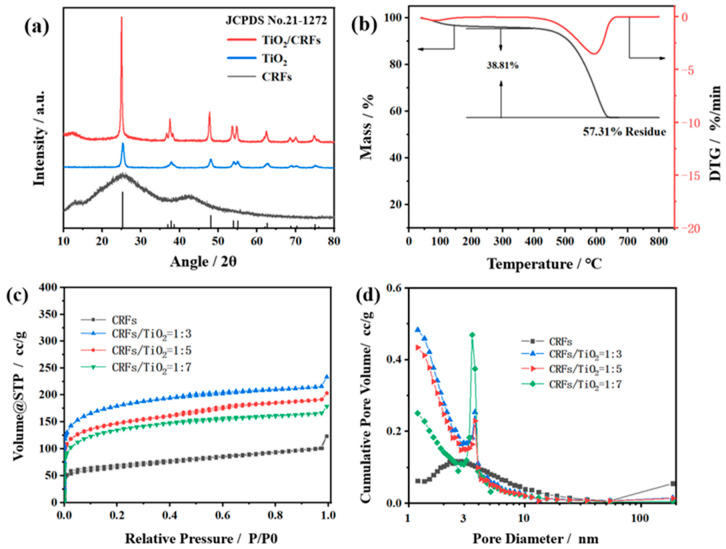
(**a**) XRD patterns of CRFs, TiO_2_ and TiO_2_/CTFs; (**b**) TG/DTG curves of TiO_2_/CRFs; (**c**) Nitrogen adsorption–desorption isotherms and (**d**) Pore size distribution for TiO_2_/CRFs.

**Figure 6 polymers-14-01593-f006:**
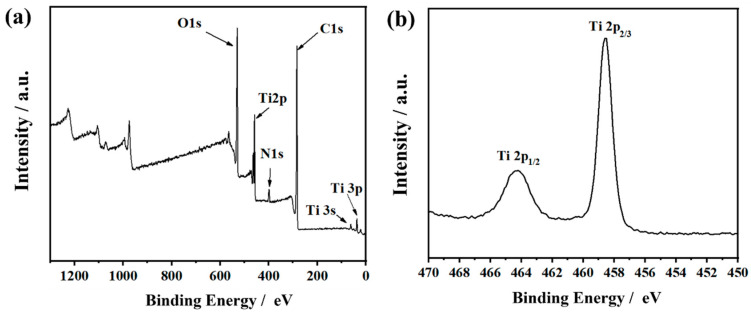
XPS spectra of (**a**) TiO_2_/CRFs and (**b**) Ti 2p.

**Figure 7 polymers-14-01593-f007:**
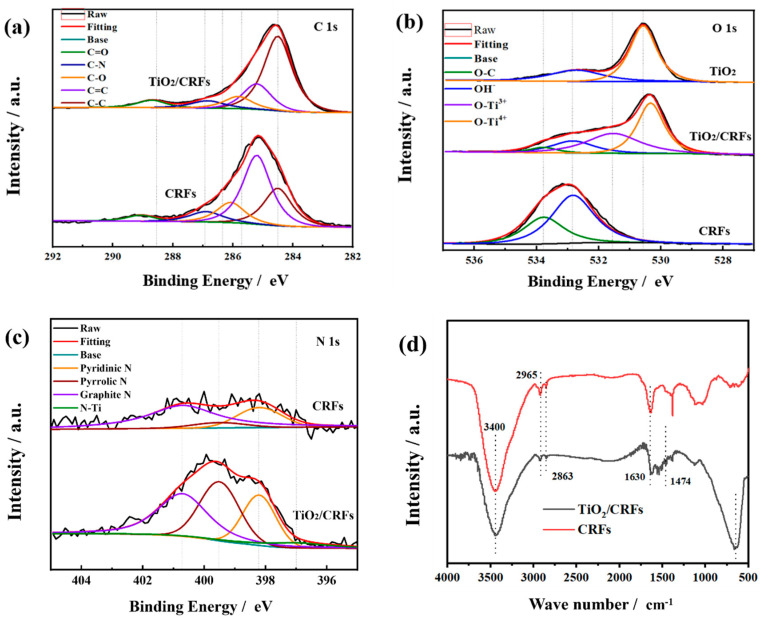
(**a**–**c**) XPS spectra of CRFs and TiO_2_/CRFs: (**a**) C 1s, (**b**) O 1s, (**c**) N 1s. (**d**) FTIR spectra of CRFs and TiO_2_/CRFs.

**Figure 8 polymers-14-01593-f008:**
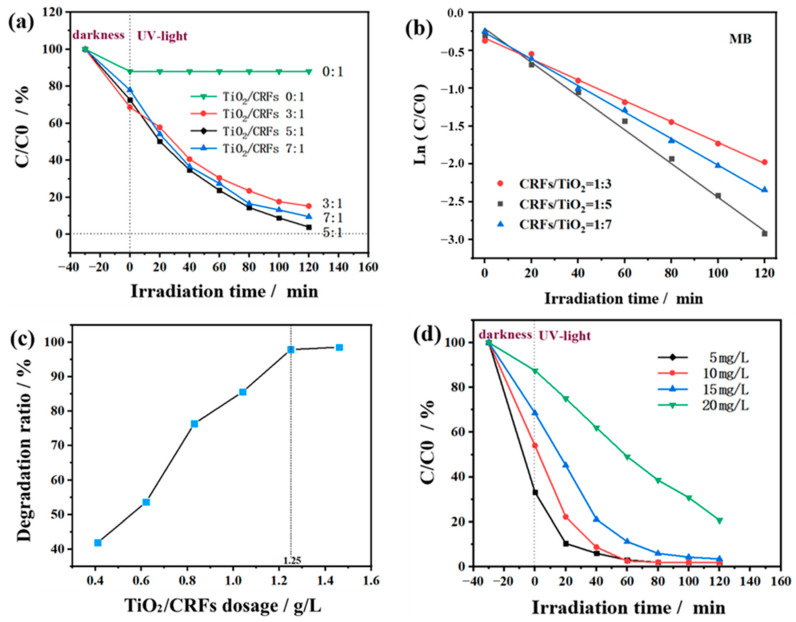
(**a**) Photocatalytic degradation activity of MB with different loads of TiO_2_/CRFs; (**b**) Linear transform ln(C/C0) = kt of the kinetic curves of dye degradation onto TiO_2_/CRFs under UV illuminations. (**c**) Effect of TiO_2_/CRFs Dosage on Degradation ratio of MB Solution; (**d**) Photocatalytic degradation activity of TiO_2_/CRFs with different MB concentrations.

**Figure 9 polymers-14-01593-f009:**
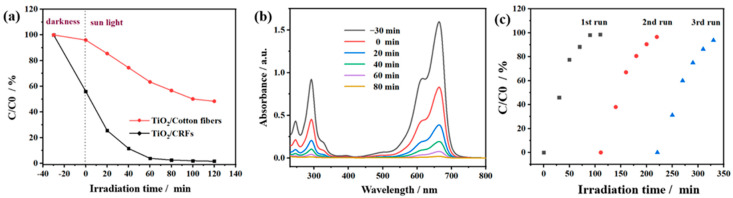
(**a**) Photocatalytic degradation activity of TiO_2_/CRFs and TiO_2_/Cotton fibers under the condition of sunlight; (**b**) Sunlight absorption curve of photodegradation of MB upon TiO_2_/CRFs; (**c**) Cycle experiments of TiO_2_/CRFs used for photocatalytic degradation of MB.

**Figure 10 polymers-14-01593-f010:**
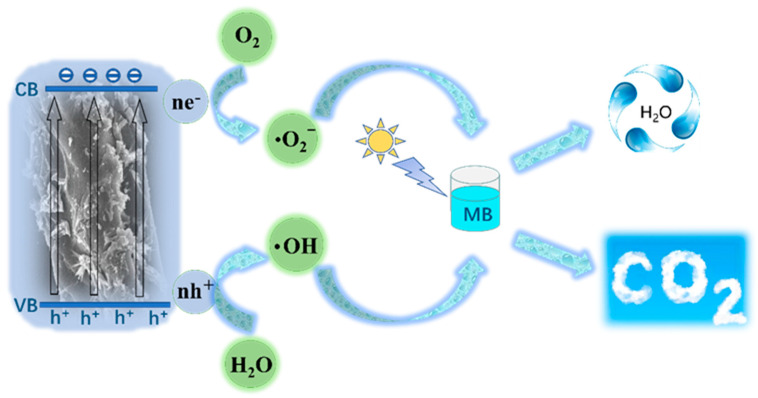
Schematic diagram of the photocatalytic mechanism for TiO_2_/CRFs photocatalyst under sunlight irradiation.

**Figure 11 polymers-14-01593-f011:**
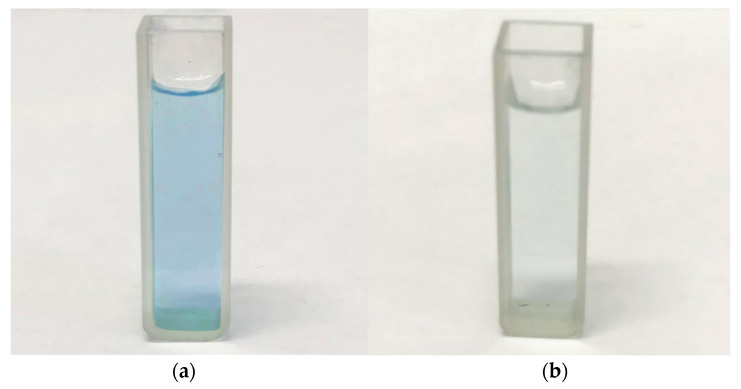
Image of (**a**) before and (**b**) after the photocatalytic degradation of methylene blue solution.

**Table 1 polymers-14-01593-t001:** The structural parameters of TiO_2_/CRFs and CRFs.

Sample	Specific Surface Area (m^2^ g^−1^)	Pore Volume (cc g^−1^)
CRFs	237.828	0.1027
TiO_2_:CRFs = 3:1	488.942	0.2099
TiO_2_:CRFs = 5:1	345.757	0.1627
TiO_2_:CRFs = 7:1	324.588	0.1580

**Table 2 polymers-14-01593-t002:** Research progress in degradation of MB dye by TiO_2_ composite catalyst.

Material	Synthesis Method	MB (mg L^−1^)	Catalyst Loading (g L^−1^)	Light Source	Highest Degradation	Cyclic Degradation	Ref.
TiO_2_	—	20	0.4	UV light	60 min, 85.45%	4 cycles, 62.49%	[40]
Bamboo biochar/TiO_2_	Calcination method	12.8	0.2	UV light	60 min, 95%	—	[51]
				Visible light	60 min, 97%	4 cycles, 75%	
SnO_2_/TiO_2_	Hydrothermal techniques	20	0.5	Visible light	50 min, 90%	4 cycles, 87%	[52]
Lignin biochar/TiO_2_	Microwave-hydrothermal and calcination method	12.8	0.2	UV light	25 min, 93%	—	[53]
CMP/TiO_2_	Sonogashira–Hagihara coupling reaction	8.6	0.28	Visible light	60 min, 96.8%	5 cycles, 93.1%	[38]
Biomolecules wrapped TiO_2_	Microwave irradiation method	10	0.2	Visible light	6 h, 90.6%	—	[54]
Hierarchical H_3_PW_12_O_40_/TiO_2_	Impregnation and layer-by-layer methods	10	0.25	UV light	5 min, 95%	5 cycles, 65% 6 cycles after calcination, 96%	[55]
P(MMA-co-BA-coMTC)/TiO_2_	Suspension polymerization	6	20	UV light	270 min, 99.66%	20 cycles, 98.7%	[56]
TiO_2_/AC	Sol–gel method	20	0.4	UV light	60 min, 99.43%	4 cycles, 88.06%	[40]
B-TiO_2_/CF	Hydrothermal method	10	0.75	UV light	10 min, 76%	5 cycles, slight decrease	[57]
				Visible light	10 min, 69%		
TiO_2_/CRFs	Carbonization, impregnation and calcination method	10	1.25	UV light	60 min, 97.9%	—	This study
				Visible light	80 min, 98.1%	5 cycles, 90.2%	

## Data Availability

Not applicable.

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
