# Peer review of "Preparing Biomass Carbon Fiber Derived from Waste Rabbit Hair as a Carrier of TiO_2_ for Photocatalytic Degradation of Methylene Blue"

_polymers, 2022, doi:10.3390/polym14081593_

Round 1

Reviewer 1 Report

This paper investigates the feasibility of using rabbit fibers as a source of C and N for doping of TiO2 catalysts and subsequent treatment of wastewater. The work is interesting and has practical applications. Although it has merit for publication, the following corrections/revisions require urgent attention before further consideration:

  1. Minor typos

line 125: “ratios” instead of “radios”

line 189: “coming” instead of “comes”

line 204: “corresponds” instead of “corresponding”

line 205: “…The results presented SHOW that….”

line 213: “Figure 5c” instead of “Figure 4c”

line 355: “nanomaterials” instead of “nanomate44rials”

line 377: “activity,” instead of “activity.”

  1. The model of TiO2 doping with C and N is rather ambiguous – No evidence of carbon doping can be found within the manuscript. The evidence of nitrogen introduction in the TiO2 lattice, in particular the XPS data, is also very unconvincing (see XPS results discussion below). From the results presented, it appears that rabbit hair fibers are decorated with TiO2 particles, with no modification of the TiO2 lattice visible. Please clarify or present more substantial arguments to convince a novice reader of these results.
  2. It is difficult to ascertain whether the carbonization step (heat treatment at 600C in N2) produces carbon-rich fibers (CRFs). EDS data of the virgin versus carbonized fibers should be compared in Fig 4 to show the effectiveness of the carbonization process.    
  3. In line 207 it is stated that, “…ratio of TiO2 into TiO2/CRFs could also reach as high as 57.31%...” How was this load-mass ratio determined?
  4. All peaks extracted from the XPS spectra should be referenced against a standard lineshape from a database. Please visit xpsfitting.com and see D. Wagner, A.V. Naumkin, A. Kraut-Vass, J.W. Allison, C.J. Powell, J.R.Jr. Rumble, NIST Standard Reference Database 20, Version 3.4 (web version) (http:/srdata.nist.gov/xps/) 2003.
  5. For example, the deconvoluted N-Ti peak of Fig 7c should be compared with a reference; Ti-N typically has a binding energy peak at 397 eV; in this article the authors identify a shallow peak at ~ 402 eV at TiN? Please clarify
  6. The relative at% of all other fitted peak data of Fig 7 should be tabulated and included. It is difficult to draw conclusions from the spectra as presented. For example, in line 241 it is stated that “…the peak value of C-C increases in TiO2/CRFs, which indicates that C=C sp2 hybridization (graphite state) is converted to C-C sp3 hybridization…” A statement of this nature should be substantiated with quantified data.
  7. The authors should include the high-res spectrum of the Ti 2p lineshape to confirm the presence or non-presence of the O-Ti-N bond; this can easily be verified against a standardized database as the O-Ti-N is a well-known vibrational state in x-ray spectroscopy.
  8. The data presented in the FTIR spectra must also be revisited – the TiN identified at ~1400 cm-1 should be verified against authenticated data. As far as known, TiN vibrational states, both in gas phase and matrix form, are detected around 1000 cm-1. See Froben and Rogge, Chem. Phys. Lett. 78 (1981) 264
  9. In line 279 the authors state that, “…concentration of MB was 10 mg L-1 (60 mL) and the sample amount of TiO2/CRFs was 50 mg…”, yet in line 306 it states that “…when TiO2/CRFs dosage is 0.41 gL-1, 1.25 gL-1, and 1.46 gL-1,…” This is inconsistent and confusing to follow. Please clarify.
  10. The authors talk about a degradation rate, yet the units of measure are in percentage (%) as shown in Fig 8c. Should this not be degradation ratio?
  11. Line 341: “…this phenomenon is due to the N-doping of TiO2/CRFs, which decreased band-gap energy of the catalyst thus enhances the solar energy absorption…” – No evidence of bandgap alteration is presented in this study; hence the authors should revisit this statement; it is speculative at best.
  12. Which “carbonaceous nanomaterials” are referred to in lines 355 onwards? No carbon nanomaterials are shown/discussed in the earlier results.
  13. Without electrochemical characterization of the catalyst, the proposed model of degradation discussed in lines 347 and further, are unsubstantiated. The authors should revisit these lines of discussion and rephrase to show that these are conclusions drawn from literature.
  14. The cycle test results of Fig 9c are also confusing – how did the authors conclude a “high level with 3.9% shrinkage observed after five consecutive tracings” from the plot of Fig 9c? Also, in Table 2 a cycle degradation of 94.2% and max degradation of 98.1% are reported. How were these values determined?

Reviewer 2 Report

The manuscript “Preparing biomass carbon fiber derived from waste rabbit hair as a carrier of TiO2 for photocatalytic degradation of methylene blue” deals with the production of TiO2/carbonized waste rabbit fibers (TiO2/CRFs) nanocomposites by a combination of carbonization, immersion, and calcination methods, using tetrabutyl titanate as the titanium source and waste rabbit hair as the carbon source. This is a hot topic that is attracting interest both in the scientific community and in the industry. Therefore, the work deserves publication; but after some revisions, as follows:

- Abstract. Add morphological properties of the nanocomposites.

- Introduction. Enlarge the state of the art on this research topic. For this purpose, see for instance this recent review: Somma et al., ChemEngineering, 2021, 5(3), 47; etc...

- R&D. Several characterizations of the nanocomposites were performed, obtaining good results. However, a photo before and after the photocatalytic degradation of methylene blue solution can be added to give a macroscopic idea of the successful procedure. The regeneration of the nanocomposites after the photocatalytic process can be approached/discussed to give higher value to the work.

- Conclusions are a summary of the work. Rewrite in a more critical way, highlighting the relevance of the findings.

- References are not in the journal style.

Round 2

Reviewer 1 Report

The authors sufficiently addressed all previous points raised. 

Manuscript may be accepted for publication. 

Author Response

We thank you very much for your positive evaluation on our paper. We will improve our future works on the evidence of carbon and nitrogen introduction in the TiO2 lattice which are suggested by the reviewer. Thank you for the comment again and wish you a successful future work!

Reviewer 2 Report

In general, the authors performed the modifications proposed by the Reviewer and improved the manuscript. However, the new references are missing, probably because the references have been not updated. Please, check.

Author Response

Dear reviewer,

We thank you very much for your valuable and very helpful comments and suggestions on our manuscript entitled “Preparing biomass carbon fiber derived from waste rabbit hair as a carrier of TiO2 for photocatalytic degradation of methylene blue” (Manuscript ID: polymers-1652206). We have modified the manuscript accordingly by updating the reference and have added the related content to the revised manuscript. We hope that you will be pleased with this revision.

Thank you and best regards.

Round 3

Reviewer 2 Report

The authors improved the manuscript.

Author Response

We thank you very much for your valuable and very helpful comments and suggestions on our manuscript. Wish you a successful future work!